# Effect of Different Immersion Tank Water Temperatures on the Microbiological Quality of Rabbit Carcasses

José Luiz Martins Silva [1], Marta Liliane de Vasconcelos [1], Joyce Graziella Oliveira [1],
Danielle de Cássia Martins da Fonseca [1], Elizangela Domenis Marino [1], Alenia Naliato Vasconcellos [1],
Luciana Oliveira Nascimento [1], Marcia Delgado da Cruz Gomes [1], Andreia Cristina Nakashima Vaz [2],
Bruna Maria Salotti de Souza [2] and Ana Maria Centola Vidal [1,*]

[1]    Faculty of Animal Science and Food Engineering, University of São Paulo, Pirassununga 13635-900, Brazil
[2]    Department of Technology and Inspection in Animal Products of the Veterinary School, Federal University of
       Minas Gerais, Belo Horizonte 31270-901, Brazil
*    Correspondence: anavidal@usp.br

**Abstract:** The pre-chilling of rabbit carcasses in an immersion tank directly interferes with microbial control. Therefore, this study was developed to examine the influence of different pre-chill tank water temperatures (4, 7, and 10 °C) on the microbiological quality of rabbit carcasses. Samples of rabbit carcasses and water and ice from the pre-chiller tank were collected; mesophilic aerobic heterotrophic microorganisms (MES), enterobacteria (EC), and coagulase-positive *Staphylococcus* (CPS) were counted; and the presence of *Salmonella* spp. was investigated. After the carcasses were immersed in the pre-chiller, there was a significant increase ($p < 0.05$) in MES (38.69 and 88.06 Log CFU·mL$^{-1}$ at 4 and 10 °C, respectively) and EC (3.20, 4.15, and 4.84 Log CFU·mL$^{-1}$ at 4, 7, and 10 °C, respectively). The average EC count tended to increase after the carcasses were immersed in the pre-chiller at different temperatures, but this increase was not significant. Water samples showed MES, EC, and CPS counts only after the immersion of the carcasses; however, the presence of these microorganisms was not detected in any of the ice samples. *Salmonella* spp. was not identified in the analyzed samples. The microorganisms analyzed at the three pre-chiller water temperatures evaluated did not multiply on the surface of the rabbit carcasses or in the pre-chiller water after carcass immersion. This study showed that none of the three pre-chilling temperatures were able to reduce the count of indicator microorganisms in the carcasses of rabbits. These data provide scientific support to discuss the need for specific norms and guidelines for rabbit meat production.

**Keywords:** enterobacteria; mesophilic; *Staphylococcus*; *Salmonella* spp.; meat; chilling





## 1. Introduction

Rabbit meat is commonly produced and consumed worldwide, but this habit is mainly found in European countries, in the Middle East, and, particularly, in China [1,2]. Rabbit meat is also appreciated by the Brazilian meat market, albeit on a smaller scale when compared with the sale of protein from other species [1–4]. Rabbit meat is prized for being high in protein and polyunsaturated fatty acids and low in fat and calories per gram, besides containing a variety of essential elements, such as minerals and vitamins [5].

Thanks to its nutritional characteristics, the use of rabbit meat is an option for human nutrition, including for consumers who seek healthy foods [3,6]. In addition, due to the estimated population growth for the coming years, the demand for food also rises. According to the Food and Agriculture Organization of the United Nations (FAO), world meat production must grow by more than 200 million tons by the year 2050. In this scenario, rabbit meat production constitutes an alternative for achieving this goal [7].

The increase in rabbit meat production is also accompanied by a growing concern about the microbiological quality of the product. Few studies exist on the microbiological

profile of rabbit carcasses, and there is no specific legislation for the production and slaughter of these animals in Brazil—lagomorphs are only mentioned briefly in a subsection of the Regulation for Industrial and Sanitary Inspection of Animal Products (RIISPOA) regarding postmortem inspection [8].

Because they are comparable in size, rabbits are slaughtered in a similar way to poultry. The rabbit slaughter process includes stunning, bleeding, skinning, rinsing, gutting, rinsing, pre-chilling, and dripping. The stage of pre-chilling by immersion is performed in both species and aims to reduce the temperature of the carcasses. This prevents possible sensory and microbiological alterations, since chilling limits the multiplication of deteriorating microorganisms and pathogens on the carcass [9,10]. When the water temperature in the chiller remains at 4 °C during the entry and exit of carcasses, MES and EC counts may decrease by up to 2 log CFU/g [11]. What differentiates both species in the chilling stage is that, in birds, it is performed in two steps, with submersion in the pre-chiller and in the main chill tank [12], whereas rabbits are only immersed in the chill tank [13].

Indicator microorganisms, such as mesophiles, *staphylococci*, and enterobacteria, can be used to assess the microbiological quality and safety of food [14]. Research must be undertaken on the microbiological quality of raw materials, and specifically regarding the animal species used, so that norms and guidelines can be established for their production and slaughter. Additionally, this information could help minimize carcass contamination through the application of hygiene procedures in the production chain.

Therefore, the objective of this study was to determine whether the use of three different water temperatures (4, 7, and 10 °C) in the pre-chiller immersion tank—used as one of the stages of the rabbit slaughter line—influences the microbiological quality of the surface of the rabbit carcass based on the count of indicator microorganisms and the presence of *Salmonella* spp. Because 7 °C is adopted as the water temperature in the immersion tank for the pre-chilling of the rabbit carcasses to control microbial load, it is believed that temperatures between 4 and 10 °C (also chilling) are also able to control the microbial load of the carcasses. In addition, this temperature range may result in lower or higher costs to the slaughterhouse and influence the quality of the meat.

## 2. Materials and Methods

### 2.1. Characteristics of the Establishment and Samples

As this study was conducted with carcasses, it received exemption from the Ethics Committee on Animal Use.

A total of 150 rabbit carcasses (12 weeks old, average weight of 1400 g, mixed-breed, males and females), along with water and ice from the pre-chiller, were collected from a packing plant located in the state of Minas Gerais, Brazil. The establishment where the study was carried out has a slaughter capacity of 500 rabbits per day and is inspected by the Federal Inspection Service (SIF). Since there is no specific legislation in Brazil for the slaughter of rabbits, the packing plant in question adopts the rules set for the slaughter of poultry as a reference to define a process adapted to the species, taking into account the differences between the species [8].

The establishment performs the carcass pre-chilling stage due to the small number of animals slaughtered per day and the short duration of the slaughter process, which takes around 90 min. Accordingly, the carcasses pass through a single pre-chill tank for immersion, which is lined with stainless steel and measures 0.6 m in height, 0.76 m in width, and 1.2 m in length, with a capacity of 547 L. The water in the tank recirculates through its own pipe and passes through coolers. However, because these are not sufficient measured taken to prevent the temperature from rising, ice produced at the same establishment is used for better temperature control.

Three different water temperatures were used during the rabbit carcass pre-chill immersion stage, namely, 4, 7, and 10 °C. Ten carcass samples were collected per temperature: five before they were immersed in the pre-chiller (SB) and five after they were immersed and remained in the tank (SA). Two water samples were also collected at all three water

temperatures tested: one before and one after immersion. Additionally, a sample of the ice that was used to control the pre-chiller temperature was collected. Collections were carried out five times throughout the years of 2020 and 2021, totaling 150 carcasses.

The carcass samples collected before and after immersion in the pre-chiller were chosen at random. The SB samples were collected during the post-evisceration rinse, a step that precedes entry into the pre-chiller.

The method used for carcass sampling was the rinsing technique, used for foods with predominantly superficial contamination. After the excess water present in the cavities resulting from the immersion was removed, each carcass was transferred to a sterile bag and subsequently weighed. Next, 0.1% peptone water was added to the bag in an amount equivalent to the weight of the carcass, and then the bag was closed and massaged. The obtained fluid was used both for the general quantification tests and presence/absence tests [15].

Two 500-mL water samples were collected from the pre-chiller in sterile Scott-type flasks through the pipe where the water recirculates—one before and one after the carcasses were immersed in the water. The external area of the pipe was sanitized with 70% ethanol, and the outflow was reduced to avoid splashing around the bottle [15].

To analyze the ice, samples were collected with sterilized spoons and transferred to sterile bags (500 g of packaged ice ready for use).

After each collection, the samples were properly identified, placed in a cooler with reusable ice, and sent to the Quali-POA Laboratory (FZEA/USP), in Pirassununga, SP, Brazil, for microbiological analyses. The distance between the plant and the laboratory was 226.3 km, and the average travel time was three hours.

### 2.2. Count of Indicator Microorganisms

A standard plate count was performed to determine the populations of mesophilic aerobic heterotrophic microorganisms (MES), coagulase-positive *Staphylococcus* (CPS), and enterobacteria (EC). For this purpose, the carcass, water, and ice samples were initially diluted in 0.1% peptone water. First, a $10^{-1}$ dilution was obtained. After homogenization, 25 mL were collected and diluted in 225 mL of 0.1% peptone water and subsequently homogenized at a low speed. The $10^{-2}$ dilution was obtained by collecting 1 mL of the $10^{-1}$ dilution and transferring it to a tube containing 9 mL of 0.1% peptone water, continuing until a $10^{-3}$ dilution was achieved [10].

For plating, 0.1 mL of each of the three dilutions was transferred to Petri dishes containing PCA (Plate Count Agar; Kasvi, Italy) for the MES count. For the CPS count, 0.1 mL of each dilution was transferred to Petri dishes containing Baird–Parker agar (Kasvi, Italy), to which 50% egg yolk emulsion and 3.5% potassium tellurite aqueous solution were added. In the case of EC, 1 mL of each dilution was transferred to sterilized Petri dishes, followed by the addition of Violet Red Bile Glucose agar (Kasvi, Italy) [16].

Plating was performed using a Drigalski loop, which was followed by incubation at 35 °C for 48 h. At the end of the established incubation period, the typical colonies were counted and multiplied by the dilution factor to obtain the result.

### 2.3. Salmonella spp. Isolation

*Salmonella* spp. was isolated by the qualitative method, which indicates the presence or absence of the microorganism in the sample [10]. The technique used was enrichment in specific broth, which was divided into two stages—pre-enrichment and selective enrichment—followed by isolation in selective media and confirmation. Pre-enrichment consisted of collecting a 25 mL sample and diluting it in 225 mL of 0.1% peptone water; then, the solution was homogenized and incubated at 35 °C for 24 h.

For selective enrichment, the flask was gently shaken, and 1 mL was transferred to 10 mL selenite cystine broth (SC) and 10 mL of tetrathionate broth (TT), followed by incubation at 42 °C for 24 h.

For differential plating, the tubes were vortexed. Using a platinum loop, one loop of TT broth was streaked onto plates containing Hektoen enteric agar (HE), bismuth sulfite agar (BS), and xylose lysine deoxycholate agar (XLD). The same procedure was repeated with SC broth. The plates were incubated at 35 °C for 24 h. After the incubation period, the development of typical colonies of *Salmonella* spp. was analyzed.

In the case of development of typical colonies of *Salmonella* spp., the lysine decarboxylase (LIA) and triple-sugar iron agar (TSI) tests were performed. The suspected culture was inoculated into the LIA and TSI tubes, and these were incubated at 37 °C for 24 h for subsequent reading.

Definitive confirmation was achieved by subjecting the typical cultures of the LIA and TSI test to specific kits (Kasvi, Italy) for *Salmonella* spp.

*2.4. Statistical Analysis*

Data were subjected to descriptive statistical analysis to evaluate the water quality in the pre-chill tank at three different temperatures (4, 7, and 10 °C) and at two different times (before and after the carcasses were immersed in the tank), which were analyzed as independent variables. The counts of mesophilic heterotrophic aerobic microorganisms, *Enterobacteriaceae* and coagulase-positive *Staphylococcus*, were analyzed as dependent variables. In the mixed linear model, temperatures and times were the fixed effects, whereas water quality varied according to the microbial count. For the analysis of water quality in the "before" time, the temperature classes of 4 °C with five replicates, 7 °C with five replicates, and 10 °C with five replicates were considered. For the "after" time, the temperature classes of 4 °C with five replicates, 7 °C with five replicates, and 10 °C with five replicates were considered. This arrangement resulted in 15 water quality classes at the three different temperatures for the "before" time, 15 water quality classes for the "after" time, 30 experimental units, and 90 observations considering the three different analyses of microbial count. The normality of the residuals was checked by the Shapiro–Wilk and Kolmogorov–Smirnov tests, and homogeneity between the variances was checked by the Levene and Bartlett tests, adopting a significance level ($\alpha$) of 0.05. Data were analyzed by ANOVA, using the MIXED procedure of Statistical Analysis System® software version 9.4 (SAS, 2018), adopting a significance level ($\alpha$) of 5% ($p < 0.05$). The statistical model below was used:

$$Y_{ij} = \mu + T_i + T^o_j + T_i \times T^o_j + e_{ij}, \tag{1}$$

where $Y_{ij}$ = observed microbial count; $\mu$ = overall mean; $T_i$ = fixed effect of time i; $T^o_j$ = fixed effect of temperature j; $T_i \times T^o_j$ = fixed interaction effect between time and temperature; and $e_{ij}$ = random error associated with each observation.

## 3. Results

*3.1. Count of Indicator Microorganisms on the Surface of the Rabbit Carcass*

The count of MES on the surface of the rabbit carcass (Tables 1 and S1) revealed a mean value of 36.29 log CFU·mL$^{-1}$ before the carcasses entered the tank at 4 °C, which did not differ significantly from the mean of 36.72 log CFU·mL$^{-1}$ found before the carcasses were immersed in the tank at 10 °C. The mean MES count of 54.40 log CFU·mL$^{-1}$ observed before the carcasses entered the tank at 7 °C differed significantly from the values detected before the carcasses entered the tank at 4 and 7 °C.

The comparison of mean counts of MES detected after the carcasses were immersed in the tank (SA) at 4, 7, and 10 °C (Tables 1 and S1) indicated a lower value at 4 °C (38.69 log CFU·mL$^{-1}$). At 7 °C, the mean MES count after immersion was 49.42 log CFU·mL$^{-1}$, which is higher than the number detected at 4 °C, but lower than the count found at 10 °C (88.06 log CFU·mL$^{-1}$).

**Table 1.** Effects of immersion of rabbit carcasses in the pre-chiller, water temperature, and their interaction on the count of mesophilic aerobic heterotrophic microorganisms (MES).

| Time | Temperature | MES (log CFU·mL$^{-1}$) | SEM | *p*-Value | | |
|---|---|---|---|---|---|---|
| | | | | Time | Temp | Time × Temp |
| SB | 4 °C | 36.29 bB | | | | |
| | 7 °C | 54.40 aB | | | | |
| | 10 °C | 36.72 bB | 9.19 | 0.0397 | 0.0404 | 0.0104 |
| SA | 4 °C | 38.69 aB | | | | |
| | 7 °C | 49.42 aB | | | | |
| | 10 °C | 88.06 aA | | | | |

SB—carcass before immersion in the pre-chiller; SA—carcass after immersion in the pre-chiller; log CFU·mL$^{-1}$—count of heterotrophic microorganisms in log colony-forming units per mL at a $10^{-1}$ dilution; SEM—standard error of the mean. Common letters correspond to means that do not differ statistically from each other at a significance level of 5% ($p < 0.05$). Different letters correspond to means that differ statistically from each other at a significance level of 5% ($p < 0.05$). Lowercase letters correspond to the comparison of means within the same temperature variable (4, 7, and 10 °C). Uppercase letters correspond to the comparison of means within the same time variable (before or after).

Considering only the temperature variable, there was a significant increase in MES count, which rose from 36.29 log CFU·mL$^{-1}$ before, to 38.69 log CFU·mL$^{-1}$ after immersion in the tank at 4 °C. There was also a significant difference between the mean MES counts of 36.72 log CFU·mL$^{-1}$ (SB) and 88.06 log CFU·mL$^{-1}$ (SA) at 10 °C. The mean EC count of 54.40 log CFU·mL$^{-1}$ did not differ significantly from the 49.42 log CFU·mL$^{-1}$ found at 7 °C, before and after immersion in the tank, respectively (Tables 1 and S1).

There was a significant interaction effect between the times (SB and SA) and the pre-chiller temperature of 10 °C on the MES count, which increased from 36.72 to 88.06 log CFU·mL$^{-1}$ (Tables 1 and S1).

The EC count (Tables 2 and S2) found before the carcasses were immersed in the tank (SB) at 4 °C was 2.38 log CFU·mL$^{-1}$. This result did not differ significantly from the mean value of 1.89 log CFU·mL$^{-1}$ observed at 7 °C, or the 2.97 log CFU·mL$^{-1}$ found at 10 °C.

**Table 2.** Effects of immersion of rabbit carcasses in the pre-chiller, water temperature, and their interaction on the mean count of enterobacteria (EC).

| Time | Treatment | EC (log CFU·mL$^{-1}$) | SEM | *p*-Value | | |
|---|---|---|---|---|---|---|
| | | | | Time | Temp | Time × Temp |
| SB | 4 °C | 2.38 aB | | | | |
| | 7 °C | 1.89 aB | | | | |
| | 10 °C | 2.97 aB | 0.8962 | 0.4361 | 0.0337 | 0.7102 |
| SA | 4 °C | 3.20 aA | | | | |
| | 7 °C | 4.15 aA | | | | |
| | 10 °C | 4.84 aA | | | | |

SB—carcass before immersion in the pre-chiller; SA—carcass after immersion in the pre-chiller; log CFU·mL$^{-1}$—count of enterobacteria in log colony-forming units per mL at a $10^{-1}$ dilution; SEM—standard error of the mean. Common letters correspond to means that do not differ statistically from each other at a significance level of 5% ($p < 0.05$). Different letters correspond to means that differ statistically from each other at a significance level of 5% ($p < 0.05$). Lowercase letters correspond to the comparison of means within the same temperature variable (4, 7, and 10 °C). Uppercase letters correspond to the comparison of means within the same time variable (before or after).

The comparison of the mean EC counts found after the carcasses were immersed in the tank (SA) at 4, 7, and 10 °C (Tables 2 and S2) shows that the lowest value (3.20 log CFU·mL$^{-1}$) occurred at 4 °C. This result did not differ significantly from the 4.15 or 4.84 log CFU·mL$^{-1}$ found after immersion in the pre-chill tank at 7 and 10 °C, respectively.

Considering only the temperature variable in isolation, the EC count increased significantly in the tank at 4 °C, rising from 2.38 log CFU·mL$^{-1}$ before immersion to 3.20 log CFU·mL$^{-1}$ after immersion. There was also a significant difference between the mean EC counts found at 7 °C (1.88 to 4.15 log CFU·mL$^{-1}$) and 10 °C (2.97 to 4.83 log CFU·mL$^{-1}$) (Table 2). There was no interaction effect between time (SB and SA) and temperature for the mean EC count (Tables 2 and S2).

In the analysis of CPS (Tables 3 and S3), before the immersion of the carcasses in the tank (SB)—here considered the time variable—the mean count found at 4 °C (62.99 log CFU·mL$^{-1}$) was higher than those detected at the temperatures of 7 and 10 °C (53.66 and 49.47 log CFU·mL$^{-1}$, respectively). Despite the difference in CPS counts between the temperatures before immersion, there was no difference in the mean CPS count after the exit of the carcasses.

**Table 3.** Effects of immersion of rabbit carcasses in the pre-chiller, water temperature, and their interaction on the mean count of coagulase-positive *Staphylococcus* (CPS).

| Time | Treatment | CPS (log CFU·mL$^{-1}$) | SEM | *p*-Value | | |
|------|-----------|-------------------------|-----|-----------|------|-------------|
| | | | | Time | Temp | Time × Temp |
| SB | 4 °C | 62.99 | | | | |
| | 7 °C | 53.66 | | | | |
| | 10 °C | 49.47 | 16.09 | 0.3918 | 0.1139 | 0.8465 |
| SA | 4 °C | 95.11 | | | | |
| | 7 °C | 68.14 | | | | |
| | 10 °C | 67.55 | | | | |

SB—carcass before immersion in the pre-chiller; SA—carcass after immersion in the pre-chiller; log CFU.mL$^{-1}$—count of coagulase-positive *Staphylococci* in log colony-forming units per mL at a 10$^{-1}$ dilution; SEM—standard error of the mean.

The samples analyzed after the immersion of the carcasses in the tank at 4 °C (SA) showed the highest mean CPS count, 95.11 log CFU·mL$^{-1}$, compared with the counts found in the SA samples at 7 and 10 °C (Tables 3 and S3). However, the mean counts observed after the carcasses passed through the pre-chill tank did not differ significantly between temperatures.

Considering only the temperature variable, there was also no significant difference in the CPS count, whose mean value of 62.99 log CFU·mL$^{-1}$ did not differ from the 95.11 log CFU·mL$^{-1}$ found at 4 °C before and after immersion, respectively. Likewise, the mean CPS counts found at 7 °C (53.66 vs. 68.14 log CFU·mL$^{-1}$) and at 10 °C (49.47 vs. 67.55 log CFU·mL$^{-1}$) did not differ significantly between the evaluation times. There was no interaction effect between time (SB and SA) and temperature (4, 7, and 10 °C) for the mean CPS count (Tables 3 and S3).

### 3.2. Count of Indicator Microorganisms in Pre-Chiller Water and Ice

The water samples collected before the immersion of the carcasses in the pre-chiller (SB) did not show microbial counts of MES, EC, or CPS at any of the three temperatures (Table 4). There were also no indicator microorganisms in the ice samples analyzed.

After the carcasses were immersed in the pre-chiller (SA), the mean MES count in the water at 4, 7, and 10 °C was 8.0, 13.0, and 4.0 log CFU·mL$^{-1}$, respectively. For EC, the observed means at the respective temperatures were 6.66, 6.07, and 7.13 log CFU·mL$^{-1}$. Finally, the mean CPS counts in the water samples after immersion of the carcasses in the tank at the temperatures of 4, 7, and 10 °C were 2.57, 7.95, and 7.12 log CFU·mL$^{-1}$, respectively.

**Table 4.** Mean population of mesophilic microorganisms (MES), enterobacteria (EC), and coagulase-positive *Staphylococcus* (CPS) in the pre-chiller water before and after immersion of the carcasses in the tank at different temperatures.

| Sample | Temperature | MES (log CFU·mL$^{-1}$) | EC (log CFU·mL$^{-1}$) | CPS (log CFU·mL$^{-1}$) |
|---|---|---|---|---|
| SB | 4 °C | 0 | 0 | 0 |
| | 7 °C | 0 | 0 | 0 |
| | 10 °C | 0 | 0 | 0 |
| SA | 4 °C | 8.0 | 6.66 | 2.57 |
| | 7 °C | 13.0 | 6.07 | 7.95 |
| | 10 °C | 4.0 | 7.13 | 7.12 |

SB—carcass before immersion in the pre-chiller; SA—carcass after immersion in the pre-chiller; log CFU·mL$^{-1}$—count of indicator microorganisms in log colony-forming units per mL at a $10^{-1}$ dilution.

*3.3. Presence of Salmonella spp.*

*Salmonella* spp. was not detected in the carcass samples collected before and after immersion in the pre-chill tank at the temperatures of 4, 7, and 10 °C. Likewise, no *Salmonella* spp. was present in the water samples before or after carcass immersion in the tank, nor in the ice samples.

**4. Discussion**

There was a significant increase in the mean MES count on the surface of the rabbit carcasses after they were immersed in the water at 4 and 10 °C. This result differs from those published by Lopes et al. (2007) [14], who did not observe a significant difference in the count of these microorganisms before and after the removal of chicken carcasses from the pre-chiller, with water temperatures ranging from 1 to 7 °C. However, the mean number of mesophiles decreased after the carcasses were immersed in the tank at 7 °C. This result was also shown in the study of Lillard (1990) [17], who evaluated the count of mesophiles before and after pre-chiller immersion in two chicken establishments and found a significant decrease, demonstrating that pre-chilling reduced surface contamination by mesophiles on chicken carcasses. Nevertheless, the study did not determine the temperature of the water in the tank.

The higher MES count obtained in the present study at the temperatures of 4 and 10 °C, before and after pre-chill immersion, suggests that the cause was the recirculation of the tank water, since there was an increase in MES at the temperature of 4 °C. It also suggests that tank water temperatures above 7 °C in a single pre-chilling step are not effective to reduce microbial count, since there was an increase in the average MES count found after the carcasses were immersed in the tank at 10 °C. This hypothesis is reinforced by the fact that MES multiply under favorable temperature conditions [18].

Enterobacteria increased significantly at all temperatures analyzed after the rabbit carcasses were immersed in the pre-chiller. This result conflicts with that observed by Lopes et al. (2007) [14], who did not find significant differences between samples of chicken carcasses before and after immersion in the pre-chiller. Souza et al. (2012) [11] reported a decrease in contamination by thermotolerant coliforms in pre-chiller water at 4 °C. SIMAS et al. (2013) [19], concluding that chilling can reduce approximately 90% of contamination by thermotolerant coliforms.

These results indicate that the contamination of the surfaces of rabbit carcasses after they were immersed in the pre-chiller increased, regardless of the temperature or quality of the water in the tank, since the water samples did not show the presence of enterobacteria before immersion. However, after immersion, these indicators were present in all treatments. This increase seen at the three temperatures can most likely be attributed to recirculation of the tank water.

Contamination by enterobacteria occurs mainly during evisceration and through cross-contamination by employees or the slaughter environment [20]. When contamination is localized and can be delimited, the affected area is condemned and the carcass is dis-



carded [8]. Nonetheless, contamination may occasionally occur in parts invisible to the employees, so the contaminated carcass may be immersed in the tank, contaminating the water and other carcasses, which may be one of the reasons for the increased contamination of carcasses and water after immersion in the tank.

Enterobacteria were found in the least quantity among all the indicators, both before and after immersion of the carcasses, which means that even in the possibility of accidental contamination, the evisceration stage and the cleaning of the equipment are efficient [15]. However, due to the inefficiency of the immersion pre-chill process regarding the control of EC, it is important to emphasize that the reduction of these microorganisms was not as expected.

Coagulase-positive *Staphylococcus* were the indicator microorganisms observed in greatest quantity, when compared with the others, both before the carcasses were immersed in the tank and afterwards. Even though the increase in CPS was not significant, the count found was high. This can be explained by the fact that carcass contamination by CPS occurs during slaughter operations, mainly during skinning, and by the hands of handlers. The problem with major CPS contamination is that this group includes *Staphylococcus aureus*, a potentially pathogenic microorganism for humans [21]. According to Kohler et al. (2008) [22] and Merz, Stephan, and Johler, (2016) [23], these are common microorganisms in the carcass of rabbits.

Some strains of *Staphylococcus aureus* cause chronic problems in rabbits, such as skin lesions, abscesses, and mastitis [24]. In the study by Ferreira, Monteiro, and Vieira-Pinto, (2014) [25], abscesses were the main cause of condemnation of rabbit carcasses, and 25.9% of these were caused by *Staphylococcus aureus*. Therefore, the step of manual skinning and the handling of carcasses by employees must be reassessed and improved to prevent large counts of CPS from contaminating the carcasses.

There was no multiplication of indicator microorganisms (MES, CPS, or EC) in the water before the carcasses entered the tank. After immersion, however, multiplication was observed in all treatments, showing that indicator microorganisms caused the contamination of the pre-chiller water after the immersion of the carcasses. The ice samples did not show counts of indicator microorganisms (MES, CPS, or EC), which means that the use of ice to control the pre-chiller water temperature did not interfere with the microbiological quality of the carcass or the water. In other words, the water and ice initially used in the pre-chill tank before the carcasses were immersed were of good quality. This reinforces the argument that the significant increase in indicator microorganisms after immersion in the pre-chiller is due to poor hygiene conditions in the environment and equipment, but mainly due to recirculation of the water in the tank.

Water recirculation is prohibited in the slaughter of poultry. Moreover, the water must be renewed according to the entry of carcasses into the tank [8]. Despite the similarity with the slaughter of poultry, the differences in care regarding the pre-chiller water may have been the reason for the greater contamination of the rabbit carcasses and the water. In the slaughter of the rabbits in question, water recirculates through pipes and is not renewed, but makeup water is added according to the entry of carcasses into the tank.

Water recirculation may be responsible for maintaining the microorganisms inside the tank for a longer time, increasing the contamination and multiplication rates. Souza et al. (2012) [11] reported that when the water is renewed and the carcasses remain in the direction opposite to the water flow (counter-flow), bacteria that are able to contaminate the carcass are mechanically removed. In addition, the MES microorganisms present in the pre-chiller water can exhibit psychrotrophic behavior and still remain in contact with the carcasses, which have the ideal intrinsic conditions for their multiplication. Coupled with this, the water passes through piping that is difficult to clean, which can favor the accumulation of dirt and the formation of biofilm by the microorganisms that were present in the tank, since they are capable of forming biofilm on stainless steel surfaces [26,27].

The non-detection of *Salmonella* spp. disagrees with results described in several studies with poultry carcasses [14,28,29]. Salmonellosis is mainly detected in poultry, cattle, and

pigs, and the main factors causing outbreaks in these species are related to characteristics of commercial farming, such as housing conditions and measures to prevent disease transmission [30]. Outbreaks of salmonellosis due to the consumption of rabbit meat are not common in Brazil, since this commercial activity is practiced on a very small scale.

Our results show that the pre-chiller water temperatures of 4, 7, and 10 °C were not efficient in improving the microbiological quality of the surface of rabbit carcasses.

## 5. Conclusions

The pre-chiller tank water temperature of 7 °C reduced the count of MES present on the surface of rabbit carcass; however, the temperatures of 4 and 10 °C did not reduce the MES count. None of the three pre-chiller tank water temperatures (4, 7, and 10 °C) reduced the count of EC present on the surface of rabbit carcasses. The increase in the population of indicator microorganisms on the carcass surface after immersion in the pre-chiller can be attributed to the non-renewal of the water. Establishing standards and guidelines for rabbit meat production can improve the quality of this food.

**Supplementary Materials:** The following supporting information can be downloaded at: https: //www.mdpi.com/article/10.3390/agriculture13020270/s1, Table S1: Effect of the temperature of the carcasses on the microbiological count MES; Table S2: Effect of the temperature of the carcasses on the microbiological count EC; Table S3: Effect of the temperature of the carcasses on the microbiological count CPS.

**Author Contributions:** Conceptualization, J.L.M.S. and M.L.d.V.; methodology, J.L.M.S., A.M.C.V., E.D.M., A.C.N.V., L.O.N., A.N.V. and M.D.d.C.G.; software, D.d.C.M.d.F. and J.G.O.; validation, A.M.C.V., A.C.N.V. and J.L.M.S.; formal analysis, D.d.C.M.d.F. and J.G.O.; investigation, J.L.M.S.; resources, J.L.M.S.; data curation, D.d.C.M.d.F. and J.G.O.; writing—original draft preparation, J.L.M.S. and M.L.d.V.; writing—review and editing, B.M.S.d.S.; visualization, B.M.S.d.S.; supervision, A.M.C.V.; project administration, A.M.C.V. All authors have read and agreed to the published version of the manuscript.

**Funding:** This research was funded by Coordenação de Aperfeiçoamento de Pessoal de Nível Superior—Brasil (CAPES) grant number [001].

**Conflicts of Interest:** The authors declare no conflict of interest.

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
