# Peer review of "Effect of Different Immersion Tank Water Temperatures on the Microbiological Quality of Rabbit Carcasses"

_agriculture, doi:10.3390/agriculture13020270_

Round 1

Reviewer 1 Report

This is an interesting paper aiming to assess the influence of different water temperatures in the pre-chill tank on the microbiological quality of rabbit carcass.

The topic investigated is of interest and fits well the overall scope of the journal. Novel and useful findings have been reported and described. Each section result easy to follow and the obtained results have been properly discussed by using updated literature.

I have not found significant inaccuracies such as to preclude the acceptance of the manuscript, but in my opinion the paper needs only a few cosmetic changes in order to further improve its quality.

Here I am reporting some suggestions:

- Title: change “carcasses” to “carcass”;

- In the Abstract, the addition of some numerical results may add value to the section;

- Add at the end of Abstract a final and brief sentence stressing the overall results and their usefulness;

- At line 37, the following reference may further support your statement https://doi.org/10.5713/ab.21.0327; and at line 43 this reference https://doi.org/10.3390/agriculture12101622

- Lines 71: It might be helpful for readers to add a few more details on rabbits (i.e. carcass weight, breed…);

- Line 92: If available, add the approval code/number;

- Tables 1-3: add the appropriate symbol for “Time*Temp”;

- The Conclusions section may be further improved.

Reviewer 2 Report

a)      About introduction section:

·         Updated references are required.

·         It is required to appropriately write the general objective of the research.

·         Formulate the hypothesis of the research.

·         It is necessary to include previous studies that describe the importance of adequate temperature (4-10 °C) in the chill tank and its effect on the growth of microorganisms.

b) About the materials and methods section:

·         It is required to describe the materials and methods used in a clear, sequential way and attach references to consult the details by readers

·         Correct citations in text.

The statistical analysis section must be included.

·         The experimental design (factorial 2x3), description of dependent variables and the statistical analysis used for all variables must be described appropriately.

·         It is necessary to describe the statistical tests used for the dependent variables, in addition to the SAS procedures.

·         What were the normality tests?

·         Why were Proc-GLIMMIX and Proc-Mixed used?

·         What were the random effects?

c) About Results and discussion section:

·         It is suggested to design 1 table with the results of the analysis of variance, and 1 table with the least-square adjusted means and their comparison.

·         It is necessary to contrast the results obtained with those reported in the literature and analyze the causes.

·         Write the conclusion based on the problem and the general objective.

In general, the manuscript requires order and synthesis of main ideas.

Round 2

Reviewer 2 Report

Statistical analysis section.

·       The statistical analysis carried out must be organized and described.

·     Initially describe the experimental design used, the dependent variables, as well as the independent ones in terms of their inclusion in the statistical model as fixed or random effects (their classes and number of observations). Subsequently The statistical tests carried out and the procedures used in the software.

·       Also, the linear expression of models used must be included (two models).

·       In case of mixed models, the structure of covariances of the random effects must be described.
